nanotechnology/organic chemistry/synthetic chemistry

palladium, magnetic, C–C coupling, heterogeneous, recyclable catalysts

**Author for correspondence:**
Rohini M. De Silva
e-mail: rohini@chem.cmb.ac.lk

[†]Present address: Department of Chemistry, University of British Columbia, Vancouver, British Columbia, Canada.
[‡]Present address: Department of Chemistry, Texas A and M University, College Station, TX, USA.

This article has been edited by the Royal Society of Chemistry, including the commissioning, peer review process and editorial aspects up to the point of acceptance.

# A magnetically retrievable air and moisture stable gold and palladium nanocatalyst for efficient C–C coupling reactions

Chatura Goonesinghe[†], Mohamed Shaik, Rivi Ratnaweera[‡], K. M. Nalin De Silva and Rohini M. De Silva

Centre for Advanced Materials and Devices (CAMD), Department of Chemistry, University of Colombo, Colombo 00300, Sri Lanka

CG, 0000-0002-0745-4641; RR, 0000-0001-7343-5695;
KMNDS, 0000-0003-3219-3233; RMDS, 0000-0003-0955-6366

In this study, we report the synthesis of a highly stable, magnetically retrievable gold and palladium nanocatalyst (AuPd@AMNPs), highly active in Suzuki cross-coupling and related homocoupling reactions. The active catalytic component in this system is palladium, which can only be stabilized in the presence of gold nanoparticles. There is no significant loss of activity even after prolonged storage exposed air and moisture. The versatile nature AuPd@AMNPs is demonstrated through the selective catalysis of the homocoupling of phenylboronic acid under low concentrations of $O_2$ and the oxidation of phenylboronic acid to phenol under high $O_2$ concentrations. AuPd@AMNPs also demonstrated Ullmann-type homocoupling of 4-iodotolene with excellent yields. The magnetically retrieved catalyst could be re-used up to six times in Suzuki–Miyaura cross-coupling with consistently high activity and with a minimal loss of the noble metal species. Through these reactions we show that the gold-stabilized AuPd@AMNPs can be used as stable and recyclable palladium reservoir for multiple palladium-catalysed reactions.

## 1. Introduction

Palladium-catalysed reactions such as C–C coupling reactions have become indispensable tools in organic chemistry [1–5]. Among these reactions, the Suzuki–Miyaura cross-coupling reaction has

been one of the most studied and extensively used palladium-catalysed reactions [1] and can be used as a benchmark to determine the versatility and reactivity of palladium catalysts. Many of these reactions have traditionally been catalysed by homogeneous catalysts. While these systems have high efficiencies and selectivity, a few major disadvantages plague their usage. In reactions catalysed by homogeneous catalysis, metal complex residues and colloidal metal particles are retained even after product separation. Most organopalladium complexes used in catalysis are air sensitive and degrade during product separation and storage; hence these expensive catalysts cannot be re-used.

Heterogeneous palladium catalysts have been developed as a potential solution to these problems [6,7]. Although a multitude of colloidal palladium catalysts with excellent activity in the Suzuki reaction have been reported [8–10] the focus of heterogeneous catalyst development has shifted towards multi-metallic nanomaterials [11,12]. Gold and palladium bimetallic systems receive special attention due to the higher catalytic activity arising from the synergistic effect of AuPd bimetallic systems. To that end, gold and palladium bimetallic nanocatalysts have been recently used in a wide array of reactions [13–17]. The oxophilicity of palladium decreases when palladium atoms are in contact with gold atoms, preventing catalyst poisoning by oxidation [18]. Furthermore, Venkatesan & Santhanalakshmi [19] propose that the increased activity is due to the sequential electronic transfers between gold and palladium in AuPd nanoclusters. It is notable that these two effects can be influenced by the catalyst carrier as well [20]. Hoshiya and co-workers show that this stabilization effect by gold can be exploited to prepare a stable palladium reservoir for Suzuki coupling reactions [21]. Nasrollahzadeh and co-workers reported an unsupported AuPd bimetallic nanocatalyst with high catalytic efficiency for several repeated cycles [22]. AuPd bimetallic systems supported on silica with superior catalytic efficiency to their monometallic counterparts in the Suzuki reaction were reported by Tan and co-workers [23]. AuPd nanoparticles supported on graphene oxide have also indicated high catalytic activity in the Suzuki reaction [24]. Another catalyst with AuPd nano-alloy decorated graphitic carbon nitride was reported to perform well in photocatalytically enhanced Suzuki coupling [25].

In addition to increased catalyst activity, the enhanced recyclability of the catalyst is also an important aspect needing attention. To this end, magnetically retrievable nanocatalysts have gained support as a catalyst carrier that can be easily separated using an external magnetic field, cutting down the number of purification, filtration and centrifugation steps required to isolate and re-use other non-magnetic catalysts. The usage of magnetically retrievable nanocatalysts in the Suzuki reaction has been steadily gaining traction with most new developments focusing on the preparation of magnetic nanoparticles with organic group functionalization that acts as ligands for various Pd or Ni complexes [26]. Even though magnetic separability increases the rate of recycling, the presence of specialized ligands requires inert conditions during reaction and storage. Another set of magnetically retrievable catalysts have been developed by immobilizing Pd nanoparticles on the surface of functionalized magnetic catalyst carriers. Because these catalysts do not use ligands they possess many of the advantages of true heterogeneous catalysis [27–30]. We propose that the marrying of high catalytic activity and stability of AuPd bimetallic systems and the recyclability of a magnetic catalyst carrier could afford a highly efficient and recyclable catalyst for the Suzuki cross-coupling and related reactions.

Herein, we report the facile synthesis and characterization of an air and moisture stable, gold- and palladium-decorated, magnetically retrievable heterogeneous nanocatalyst with excellent catalytic activity in the Suzuki–Miyaura cross-coupling reaction. Furthermore, we show the versality of this catalytic system though its application in homocoupling, oxidation and hydrogenation reactions. Finally, we show the robust nature of this system through the stable catalyst activity, morphology and composition after repeated usage and prolonged storage exposed to air and moisture.

# 2. Results and discussion

## 2.1. Synthesis and characterization of AuPd@AMNPs

Gold- and palladium-decorated amine-functionalized nanoparticles (AuPd@AMNPs) are synthesized using a facile two-step procedure. This involves the single-step synthesis of an amine-functionalized magnetite core using an ethylene glycol-based solvothermal synthesis [29] followed by the use of simple chloride salts of gold and palladium as precursors for surface decoration of AMNPs. The *in situ* rapid reduction of $Au^{3+}$ and $Pd^{2+}$ using sodium borohydride yields the final catalyst used in further experiments. This process and the chemical reactions involved (equation (2.1)) are presented in

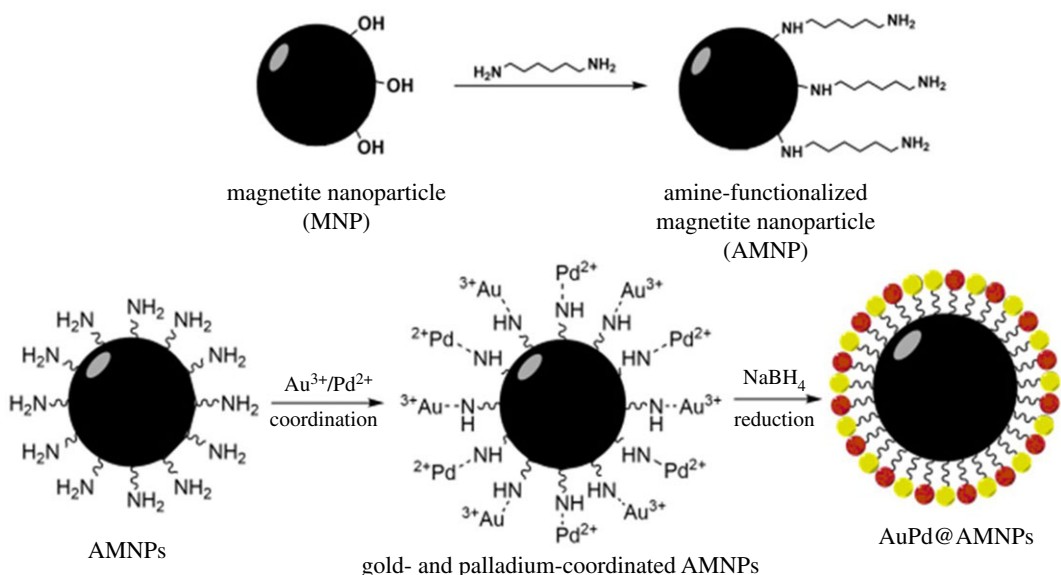

**Scheme 1.** Preparation of amine-functionalized iron oxide nanoparticles (AMNPs) followed by the surface decoration by a bimetallic nano-alloy (AuPd@AMNPs).

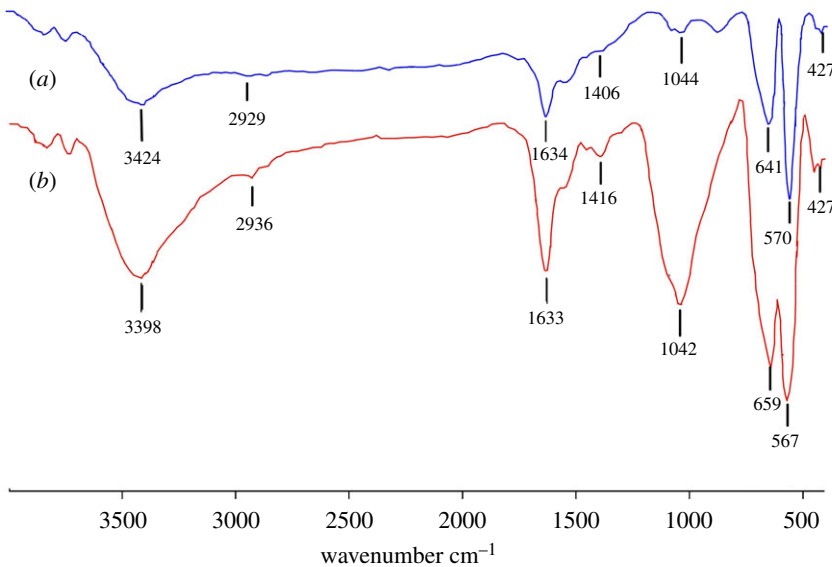

**Figure 1.** FTIR spectra of (a) AMNPs prior to AuPd decoration and (b) washed and dried AuPd@AMNPs after usage in Suzuki coupling.

scheme 1. The synthesis is kept simple with the usage of a minimum number of readily available precursors and reagents.

$$\left. \begin{array}{rcl} Fe^{3+} \ + \ 3OH^- &\rightarrow& Fe(OH)_3 \\ 10\,Fe(OH)_3 \ + \ C_2H_6O_2 &\rightarrow& 2\,H_2CO_3 \ + 6\,H_2O \ + 10\,Fe(OH)_2 \\ 2\,Fe(OH)_3 \ + \ Fe(OH)_2 &\rightarrow& Fe_3O_4 \ + \ 4\,H_2O \end{array} \right\} \tag{2.1}$$

The Fourier transform infrared (FTIR) spectrum (figure 1a) of the synthesized AMNPs shows strong absorption bands at 427 and 570 cm$^{-1}$ corresponding to Fe–O and Fe–O–Fe stretching vibrational modes characteristic to magnetite. Confirming the synthesis of small-sized magnetite nanoparticles, the band at 641 cm$^{-1}$ arises from the splitting of the Fe–O band at 570 cm$^{-1}$ of bulk magnetite. The bands at 3424 and 1634 cm$^{-1}$ correspond to N–H stretching and bending modes of the free –NH$_2$ group. The band at 1044 cm$^{-1}$ represents the C–N stretching vibration. The bands at 1406 and 2929 cm$^{-1}$ correspond to

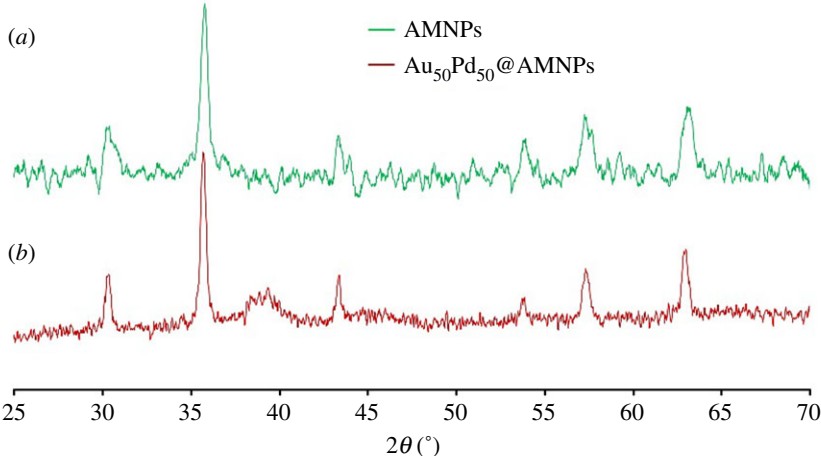

**Figure 2.** XRD patterns of (*a*) neat AMNPs and (*b*) Au$_{50}$Pd$_{50}$@AMNPs.

C–H symmetric bending and stretching modes of the –CH$_2$– groups of the hexyl moiety from 1,6-diaminohexane. These results confirm the synthesis of small AMNPs [31].

The formation of AuPd@AMNPs and the determination of the crystalline phase of the structures was performed using powder X-ray diffraction (XRD). Figure 2 shows the XRD patterns obtained for undecorated AMNPs and AuPd@AMNPs. The XRD peaks observed at $2\theta = 31.1°$, 35.5°, 43.1°, 57.3° and 63.1° are indexed to the (220), (311), (400), (422), (500) and (440) planes of fcc Fe$_3$O$_4$ (JCPDS 751610). The high level of crystallinity of the AMNPs is confirmed by the sharp characteristic Fe$_3$O$_4$ peaks. Apart from the Fe$_3$O$_4$ peaks, none of the characteristic gold or palladium peaks are clearly observable in the XRD pattern obtained for AuPd@AMNPs (figure 2*b*). However, in the $2\theta = 37°–40°$ region a low intensity broad amorphous halo is detected. This type of amorphous halo is usually detected in the formation of irregular metal alloys [32]. We propose that this broad region is the result of the lowered $2\theta$ values of palladium (111) and the gold (111) reflections overlapping upon the formation of an amorphous alloy on the surface of the AMNPs through reduction of Au$^{3+}$ and Pd$^{2+}$. The low intensity of this peak is attributed to the small particle size due to rapid reduction as well as the low concentrations of gold and palladium.

Microscopic characterization of the catalyst was done at each synthetic step using scanning electron microscopy (SEM). Concurrent surface elemental analysis by energy dispersive spectroscopy (EDS) was conducted to confirm the surface decoration of AMNPs by gold and palladium. SEM of the newly synthesized AMNPs (figure 3*a*) shows monodisperse irregular spherical AMNP particles with a diameter of approximately 45 nm. Surface elemental analysis by EDS only shows iron, oxygen and carbon in significant amounts, as expected (electronic supplementary material, figures S1 and S2). Upon reduction of the gold and palladium precursors, new surface features with an approximately 10 nm diameter is seen on the surface of the AMNPs (figure 3*b*). EDS analysis indicated that these new structures are gold and palladium clusters decorating the surface of AMNPs (electronic supplementary material, figures S3 and S4). While some particle aggregation is visible, discrete AMNP particles are still discernible.

During synthesis of AuPd@AMNPs, the total amount of gold and palladium is kept constant and is added in predetermined ratios on to the AMNPs. Elemental analysis using inductively coupled plasma mass spectrometry (ICPMS) indicates that metal loading does not follow the initial ratio of precursors (table 1). Gold shows a more efficient loading compared to palladium even at higher Pd$^{2+}$ concentrations (table 1, entries 2, 3). When gold is not present only 10% of the initial palladium added decorates the surface of the AMNPs. While this number increases with the addition of 25% gold in Au$_{25}$Pd$_{75}$@AMNPs, however, over 75% of palladium added is not deposited (electronic supplementary material, table S1). In addition to the low metal loading, both Au$_0$Pd$_{100}$@AMNPs and Au$_{25}$Pd$_{75}$@AMNPs cannot be efficiently magnetically separated as they undergo rapid oxidation upon exposure to air. For this reason, these two systems are not used for further catalytic studies. Both metals showed the highest rates of loading in the Au$_{50}$Pd$_{50}$@AMNPs system (1.19% Pd w/w and 3.74% Au w/w). Based on these observations we conclude that gold stabilizes the palladium atoms on AMNPs preventing its oxidation while assisting the precipitation of palladium. This behaviour is in agreement with previous studies that concluded that this effect is due to similar lattice constants between gold and palladium in addition to the ability of gold atoms to withdraw electrons from neighbouring metal atoms [33,34].

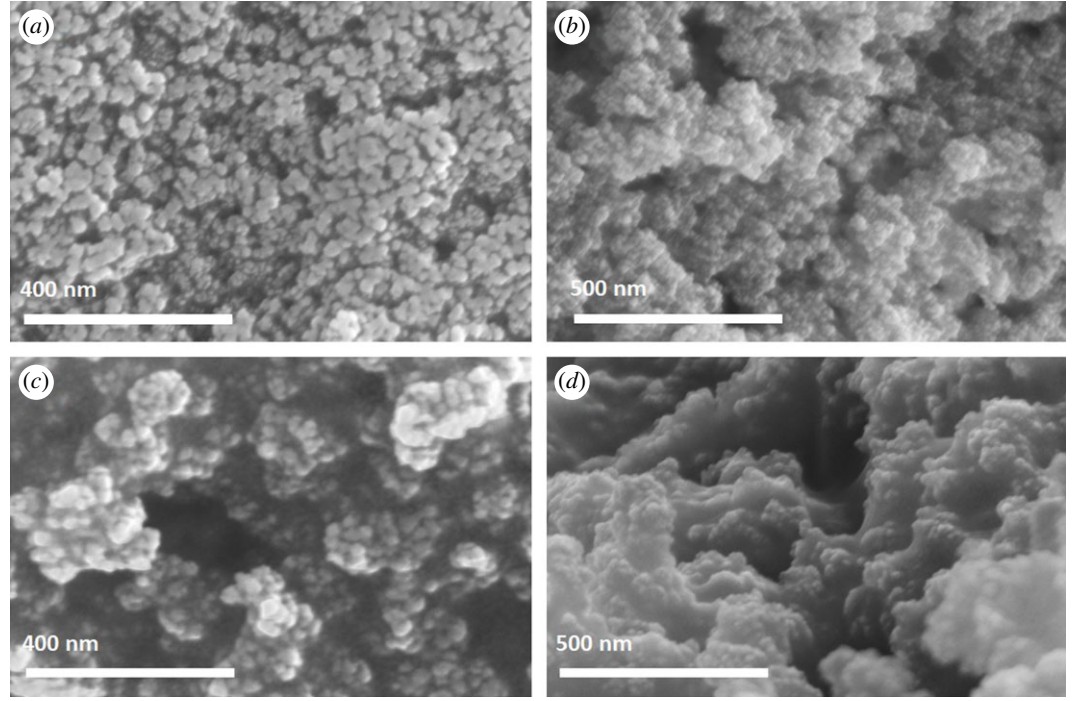

**Figure 3.** SEM images of (*a*) AMNPs (*b*) AuPd@AMNPs (*c*) AuPd@AMNPs after a single use and (*d*) AuPd@AMNPs after six cycles.

**Table 1.** Analysis of gold and palladium loading on AMNPs.

| | catalyst | initial mole %[a] | | atom %[b] | | loading %[c] | |
|---|---|---|---|---|---|---|---|
| | | Au | Pd | Au | Pd | Au | Pd |
| 1 | AMNP | 00 | 00 | 00 | 00 | 00 | 00 |
| 2 | Au$_0$Pd$_{100}$@AMNP | 00 | 100 | 0 | 100 | 00 | 10 |
| 3 | Au$_{25}$Pd$_{75}$@AMNP | 25 | 75 | 35 | 65 | 39 | 24 |
| 4 | Au$_{50}$Pd$_{50}$@AMNP | 50 | 50 | 67 | 33 | 57 | 34 |
| 5 | Au$_{75}$Pd$_{25}$@AMNP | 75 | 25 | 89 | 11 | 56 | 21 |
| 6 | Au$_{90}$Pd$_{10}$@AMNP | 90 | 10 | 95 | 05 | 44 | 19 |
| 7 | Au$_{100}$Pd$_0$@AMNP | 100 | 00 | 100 | 00 | 47 | 00 |

[a]Total $Au^{3+}$ + $Pd^{2+}$ amount = 0.67 mmol per 1 g of AMNPs.
[b]Determined by ICPMS.
[c](mmol of actual Au or Pd/mmol of initial Au or Pd) × 100.

## 2.2. Application of AuPd@AMNPs to Suzuki cross-coupling

In order to determine the reactivity of AuPd@AMNPs towards Suzuki cross-coupling, the coupling between 4-iodotoluene and phenylboronic acid resulting in 4-phenyltoluene is used as a model reaction. The optimal composition of the catalyst was determined using AuPd@AMNPs with varied gold and palladium loadings under identical conditions. Reactions were initially conducted in tetrahydrofuran (THF)/H$_2$O (2 : 1) with Na$_2$CO$_3$ as a mild base. Although yields are improved with a stronger base, all systems proceed to furnish low yields (table 2, entries 1–6). The active catalytic phase in reactions involving colloidal palladium has been indicated to be leached palladium atoms [35]. We propose that these low yields are due to the low dispersal of both phenylboronate ions [36] and the active catalytic phase in the mildly polar THF, [37] resulting in poor interaction between coupling partners and the active catalytic phase. It is notable that the self-coupled product of the aryl halide is also detected in addition to the cross-coupling product. We propose that competitive Ullmann-type homocoupling occurs on gold and palladium nanoparticles under these reaction

**Table 2.** Test for catalytic activity of AuPd@AMNPs in the Suzuki–Miyaura cross-coupling reaction. Reaction conditions: Phenylboronic acid (2.0 mmol) and 4-iodotoluene (2.0 mmol) in solvent (10 ml), AuPd@AMNPs (100 mg), 5 ml of base (10% w/v), reflux under $N_2$ for 4 h.

| | catalyst | solvent | base | C–C yield%[a] | H–C yield%[b] |
|---|---|---|---|---|---|
| 1 | $Au_0Pd_0$@AMNP | $THF/H_2O$ | $Na_2CO_3$ | 0 | 0 |
| 2 | $Au_{100}Pd_0$@AMNP | $THF/H_2O$ | $Na_2CO_3$ | 2 | 2 |
| 3 | $Au_{90}Pd_{10}$@AMNP | $THF/H_2O$ | $Na_2CO_3$ | 3 | 11 |
| 4 | $Au_{75}Pd_{25}$@AMNP | $THF/H_2O$ | $Na_2CO_3$ | 10 | 5 |
| 5 | $Au_{50}Pd_{50}$@AMNP | $THF/H_2O$ | $Na_2CO_3$ | 31 | 2 |
| 6 | $Au_{50}Pd_{50}$@AMNP | $THF/H_2O$ | $Na_3PO_4$ | 45 | 8 |
| 7 | $Au_0Pd_0$@AMNP | $ACN/H_2O$ | $Na_2CO_3$ | 0 | 0 |
| 8 | $Au_{100}Pd_0$@AMNP | $ACN/H_2O$ | $Na_2CO_3$ | 1 | 1 |
| 9 | $Au_{90}Pd_{10}$@AMNP | $ACN/H_2O$ | $Na_2CO_3$ | 96 | 2 |
| 10 | $Au_{75}Pd_{25}$@AMNP | $ACN/H_2O$ | $Na_2CO_3$ | 96 | 2 |
| 11 | $Au_{50}Pd_{50}$@AMNP | $ACN/H_2O$ | $Na_2CO_3$ | 98 | 1 |
| 12 | $Au_{50}Pd_{50}$@AMNP | $ACN/H_2O$ | $Na_2PO_4$ | <99 | 0 |

[a]Cross-coupling product determined by GC–MS.
[b]Homocoupling product determined by GC–MS.

**Table 3.** Role of the linker arm in the catalytic activity of AuPd@AMNPs. Reaction conditions: phenylboronic acid (2.0 mmol) and 4-iodotoluene (2.0 mmol) in ACN (10 ml), $Au_{50}Pd_{50}$@AMNPs (100 mg), 5 ml of $Na_3PO_4$ (10% w/v), reflux under $N_2$ for 4 h.

| | linker | yield %[a] |
|---|---|---|
| 1 | 1,6-diaminohexane | 98 |
| 2 | 1,3-diaminopropane | 85 |

[a]Determined by GC–MS.

conditions [38–40]. In order to increase the dispersal of the catalyst, the same set of reactions was carried out in the more polar acetonitrile [37] (ACN)/$H_2O$ (2:1) medium (table 2, entries 7–12). With the change in reaction conditions, both catalytic activity and the selectivity increases in all palladium-containing systems. Based on these results, it is evident that monometallic gold does not act as a catalyst in this context. Keeping with these observations, the $Au_{50}Pd_{50}$@AMNP catalyst is used with ACN/$H_2O$ (2:1) solvent system and $Na_3PO_4$ as base for further experiments.

Another factor that was evaluated is the length of the diaminoalkane linker arm used to anchor the gold and palladium clusters to the surface of the magnetite particles. The results, presented in table 3, show that the usage of the longer 1,6-diaminohexane as the linker arm provides greater yields.

The optimized reaction conditions are used to evaluate the coupling of various aryl halides to phenylboronic acid to determine the catalytic activity of $Au_{50}Pd_{50}$@AMNPs (table 4). A general observation is that in the reactions of aryl halides bearing electron-withdrawing substituents (–$CF_3$, –F) and weakly electron-donating (–$CH_3$) substituents, $Au_{50}Pd_{50}$@AMNPs acts as an excellent catalyst giving consistently high yields. However, when the aryl halide coupling partner bears a strongly electron-donating substituent (–$OCH_3$, –OH) the yields are drastically reduced. In the coupling of 4-bromophenol with phenylboronic acid to yield 1,1′-biphenyl-4-ol, the product yield increases from 8% to 34% when NaOH is used as the base. This indicates that stronger bases can be used to overcome the sluggish reactivity of aryl halides with strong electron donor substituents.

**Table 4.** Suzuki–Miyaura reaction of aromatic aryl halides and phenylboronic acid catalysed by $Au_{50}Pd_{50}$@AMNPs. Reaction conditions: phenylboronic acid (2.0 mmol) and arylhalide (2.0 mmol) in ACN (10 ml), $Au_{50}Pd_{50}$@AMNPs (100 mg), 5 ml of $Na_3PO_4$ (10% w/v), reflux under $N_2$ for 4 h.

| | aryl halide | product | yield %[a] |
|---|---|---|---|
| 1 | | | 98 |
| 2 | | | 99 |
| 3 | | | 99 |
| 4 | | | 10 |
| 5 | | | 8<br>34[b] |

[a]Determined by GC–MS.
[b]NaOH used as base.

**Table 5.** Other reactions catalysed by $Au_{50}Pd_{50}$@AMNPs.

| | reaction | yield %[e] |
|---|---|---|
| 1[a] | | >99 |
| 2[b] | | >99 |
| 3[c] | | >99 |
| 4[d] | | >99 |

Reaction conditions: [a]phenylboronic acid (2.0 mmol), ethanol (10 ml), $Au_{50}Pd_{50}$@AMNPs (100 mg), 5 ml of $Na_2CO_3$ (10% w/v), reflux in open to air for 4 h.
[b]Phenylboronic acid (2.0 mmol), ethanol (10 ml), $Au_{50}Pd_{50}$@AMNPs (100 mg), 5 ml of $Na_2CO_3$ (10% w/v), reflux under $O_2$ for 4 h.
[c]4-iodotoluene (2.0 mmol), ACN (10 ml), $Au_{50}Pd_{50}$@AMNPs (100 mg), 5 ml of $Na_3PO_4$ (10% w/v), reflux under $N_2$ for 4 h.
[d]1-chloro-3-nitrobenzene (2.0 mmol), ethanol (10 ml), $Au_{50}Pd_{50}$@AMNPs (100 mg), reflux under $H_2$ for 4 h.
[e]Determined by GC–MS.

## 2.3. Application of AuPd@AMNPs to other reactions

The formation of biphenyl and phenol in some Suzuki coupling experiments partially exposed to air led us to investigate the activity of $Au_{50}Pd_{50}$@AMNPs in catalysing homocoupling and oxidation of phenylboronic acid. The homocoupling reaction of phenylboronic acid resulting in the formation of biphenyl has been reported to be catalysed in the presence of oxygen by both gold and palladium nanoparticle as well as gold/palladium nano-cluster species, with phenolic species being a frequent side-product of this reaction [41–45]. We managed to isolate these two reactions by changing the reaction conditions. By refluxing the

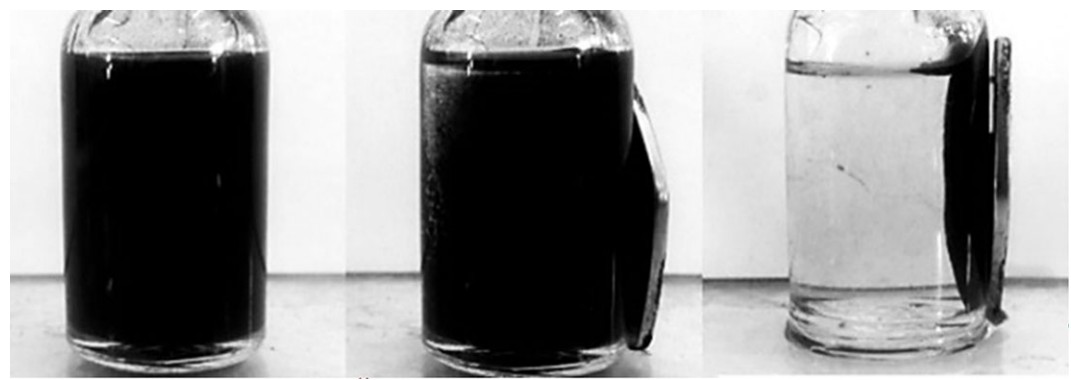

**Figure 4.** Recovery of AuPd@AMNPs at the end of a reaction cycle using an external magnet.

**Table 6.** Catalyst recyclability of $Au_{50}Pd_{50}$@AMNPs in the Suzuki–Miyaura cross-coupling reaction.

| run | 1 | 2 | 3 | 4 | 5 | 6[a] |
|---|---|---|---|---|---|---|
| yield %[b] | 99 | 95 | 96 | 97 | 97 | 95 |

Reaction conditions: phenylboronic acid (2.0 mmol) and 4-iodotoluene (2.0 mmol) in ACN (10 ml), $Au_{50}Pd_{50}$@AMNPs (100 mg), 5 ml of $Na_3PO_4$ (10% w/v), reflux under $N_2$ for 4 h.
[a]Catalyst reused after three months of storage in ambient conditions.
[b]Determined by GC–MS.

reaction mixture in under air, i.e. low concentrations of $O_2$, biphenyl is produced without any phenolic side-product (table 5, entry 1). However, when refluxed under pure $O_2$ phenylboronic acid is selectively oxidized to produce phenol (table 5, entry 2). The mechanism involved in this $O_2$ concentration-dependent switching is unclear. It is possible that the hydroxylation of phenylboronic acid involves the formation of peroxides under high $O_2$ concentrations [46]. However, further mechanistic studies will have to be conducted to fully understand this difference in reactivity. In lower polarity, solvents where the boronate intermediate species of Suzuki coupling are unstable, Ullmann homocoupling reaction of 4-iodotoluene is observed as a competing reaction. This type of homocoupling has been previously reported to be catalysed by gold and palladium nanoparticles [38–40,47]. In the absence phenylboronic acid, 4-iodotoluene undergoes this type of homocoupling reaction, giving very high yields (table 5, entry 3). Considering the wide use of palladium-catalysed hydrogenation reactions, [30,48] we demonstrate the application of $Au_{50}Pd_{50}$@AMNPs in the hydrogenation of 1-chloro-3-nitrobenzene to 3-chloroaniline under a $H_2$ atmosphere (table 5, entry 4). These reactions serve to demonstrate the versatility of $Au_{50}Pd_{50}$@AMNPs in catalysing a range of gold- and palladium-catalysed reactions.

## 2.4. Recyclability of AuPd@AMNPs

The recyclability of the catalyst is evaluated by re-using it in consecutive Suzuki cross-couplings between 4-iodotoluene and phenylboronic acid. This is done by recovering the particles using an external magnet (figure 4) and re-using the washed catalyst in the next reaction cycle. As the catalyst support is composed of magnetic AMNPs, separation steps such as centrifugation and filtration are not required. The first five cycles (table 6, entries 1–5) were conducted two weeks after the initial synthesis of the catalyst. The $Au_{50}Pd_{50}$@AMNPs are stored under ambient conditions without further precautions (mean RT 27°C and RH 80%). All five cycles gave very high yields (greater than 95%) with minimal loss of activity. The sixth cycle (table 6, entry 6) was conducted after a further three months had elapsed from cycle five ($Au_{50}Pd_{50}$@AMNPs were stored under ambient conditions). This cycle also shows a high yield (95%) with little loss of activity. The facile magnetic separation, the efficient recyclability and the non-rigorous long-term storage conditions makes the $Au_{50}Pd_{50}$@AMNP catalyst an excellent catalyst for the Suzuki cross-coupling reaction.

## 2.5. Post-reaction characterization of the catalyst

Post-reaction analysis of the $Au_{50}Pd_{50}$@AMNPs catalyst shows little loss of the active catalytic phase with only minor changes in the surface morphology of the catalyst. FTIR spectrum of post-reaction

$Au_{50}Pd_{50}$@AMNPs does not show a significant difference from pre-use catalyst, indicating the AMNP support structure does not undergo surface structural changes during catalyst usage (figure 1*b*). The increased intensity of peaks around 1040 and 3400 cm$^{-1}$ are attributed to residual ethanol from catalyst washing after the reaction. SEM imaging of $Au_{50}Pd_{50}$@AMNPs after a single usage show some particle aggregation (figure 3*c*). Aggregation is significantly pronounced at the end of six cycles of usage (figure 3*d*). However, it is notable that particle aggregation does not influence the catalytic activity of $Au_{50}Pd_{50}$@AMNPs. EDS analysis shows the gold/palladium nanoclusters decorating the surface of the AMNPs mostly remain intact even after usage (electronic supplementary material, figures S5 and S6). Elemental analysis using ICPMS shows that after a single usage there is negligible loss of gold (3.74% w/w) while 3.5 mol% of palladium (1.15% w/w) is lost. After usage in six reaction cycles we found a 23 mol% loss of gold (2.88% w/w) and a 17 mol% loss of palladium (0.99% w/w) relative to the initial catalyst (electronic supplementary material, table S1). While palladium leaching is significant at the end of six catalytic cycles, the relative stability of noble metal content and the very small loss of palladium at the end each reaction cycle points to this system acting as a stable palladium reservoir over multiple catalytic cycles. The structural and compositional analysis of the post-reaction catalyst clearly show the highly stable nature of $Au_{50}Pd_{50}$@AMNPs with minimal loss of the active catalytic phase even at the end of several catalytic usages and prolonged storage under ambient conditions.

# 3. Conclusion

In this study, we synthesized a nanoscale catalyst comprising AuPd clusters decorated on the surface of amine-functionalized magnetite nanoparticles (AuPd@AMNPs). Gold does not play a catalytic role. However, gold atoms play the dual role of preventing catalyst poisoning through oxidation and enhancing reactivity though synergistic effects. Thereby, this catalyst is highly air and moisture stable in usage and storage. The catalyst demonstrated excellent reactivity and recyclability in the Suzuki–Miyaura cross-coupling reaction. The versatility of this catalyst is demonstrated though its high activity in the homocoupling and oxidation of phenylboronic acid, the Ullmann homocoupling of 4-iodotoluene and hydrogenation of a nitroaromatic compound. We aim to use this catalytic system to further study other reactions involving palladium catalysis.

# 4. Experimental section

## 4.1. General considerations

The synthesized nanoparticles were analysed using Fourier transform infrared (FTIR) spectroscopy using a Bruker Vertex80 spectrophotometer in the wavenumber range 400 and 4000 cm$^{-1}$. Samples were prepared in the form of pellets using KBr, maintaining the KBr:sample mass ratio at 1:10. Scanning electron microscopy (SEM) images and surface elemental analysis of AMNPs and AuPd@AMNPs was conducted using a Hitachi SU6600 FE-SEM. X-ray diffraction analysis of the synthesized AMNPs and AuPd@AMNPs was performed using a Bruker D8 Focus X-ray powder diffractometer using Cu K$\alpha$ radiation (=0.154 nm) over the 2$\theta$ range of 5°–80°, with a step size of 0.02° and a step time of 1 s. Elemental analysis of AMNPs and AuPd@AMNPs was done using inductively coupled plasma mass spectrometry (ICPMS) using the Agilent 7000 ICPMS System. Samples: 25.0 mg were digested in $H_2O_2$ followed by dissolution in a mixture of HCl and $HNO_3$, the final volume was made up to 10.0 ml. The products of catalytic reactions were characterized by GC–MS using Agilent 5977A Series GC/MSD System having a 30 m × 250 µm × 0.25 µm HP-5 ms column. The temperature programme was 40–250°C at 30°C min$^{-1}$ with a final temperature isothermal hold for 12 min. The MS mass limit was set between 50 and 450 Da.

## 4.2. Materials

All the chemicals were commercially obtained in analytical grade and used without further purification. Gold- and palladium-decorated amine-functionalized magnetite nanoparticles was synthesized by using iron(III) chloride ($FeCl_3$, anhydrous, greater than 99% w/w, Merck Ltd), 1,6-hexanediamine (The British Drug House Ltd), sodium acetate anhydrous (Daejung Chemicals & Metals Co.), ethylene glycol (anhydrous 99.8%, Sigma-Aldrich), sodium borohydride (Daejung Chemicals & Metals Co.), palladium(II) chloride ($PdCl_2$, 99.9%, metal basis Pd 59.0% min., Alfa Aesar) and gold chloride

hydrate (HAuCl$_4$, purum, 50% Au, Fluka). Distilled water was used for preparation of all aqueous solutions. The cross-coupling reactions were conducted using phenylboronic acid (Sisco Research Laboratories Pvt. Ltd) and organohalides such as 4-iodotoluene (Alfa Aesar, UK), 4-bromoanisole (greater than 97%, Fluka), 1-bromo-4-fluorobenzene (greater than 97%, Fluka), 4-bromophenol (BDH Chemicals Ltd), 3-bromobenzotrifluoride (greater than 97%, Fluka). Sodium orthophosphate (greater than or equal to 96%, Sigma-Aldrich) was used as the base in the reaction. Acetonitrile (greater than or equal to 99.8%, Sigma-Aldrich) and diethyl ether (greater than or equal to 99.8%, Sigma-Aldrich) were used as solvents.

## 4.3. Synthesis of AMNPs

AMNPs were synthesized using a previously reported solvothermal technique [23]. A mass of 2.0 g FeCl$_3$, 4.0 g anhydrous sodium acetate and 7.8 ml of 1,6-diaminohexane in 50 ml ethylene glycol was stirred at 45°C for 30 min. This homogenized solution was transferred to a steel bomb reactor and maintained at 200°C for 8 h. The synthesized AMNPs were thoroughly washed with distilled ethanol several times to remove solvent and unreacted 1,6-diaminohexane. The AMNPs were vacuum dried at 45°C to obtain a black powder.

## 4.4. Synthesis of Au$_{50}$Pd$_{50}$@AMNPs

The synthesized AMNPs sample (1.0 g) was dispersed in 50 ml ethanol and ultrasonicated for 30 min. To this suspension, HAuCl$_4$ (0.33 mmol in solution) and PdCl$_2$ (0.33 mmol in solution) was added and the resulting mixture was ultrasonicated again for 60 min. Following ultrasonication, the system was flushed using gaseous nitrogen and the solution was stirred under a nitrogen atmosphere for 20 min. Next, an excess of cold, 0.01 M sodium borohydride was allowed to drip down slowly on to the mixture with vigorous stirring, under a nitrogen atmosphere. The mixture was stirred for two hours to obtain the reduced products, which were separated out using an external magnet. The Pd/Au@AMNPs were thoroughly washed using distilled ethanol.

## 4.5. Representative Suzuki–Miyaura cross-coupling reaction

In a round-bottomed flask, phenylboronic acid (2.0 mmol) and arylhalide (2.0 mmol) were mixed with 10 ml ACN and stirred for 15 min until all solids dissolved. To this mixture, 5.0 ml of Na$_3$PO$_4$ (10% w/v) was added followed by 100 mg of Au$_{50}$Pd$_{50}$@AMNPs. The nanocatalyst was suspended in solution using ultrasonication for 30 min. The solution was heated to reflux and reacted for 4 h under a N$_2$ balloon. The reaction was monitored through thin layer chromatography (TLC) with *n*-pentane used as the eluting solvent. The mixture was cooled to room temperature and diluted with distilled water (5 ml) and diethyl ether (20 ml). The catalyst was magnetically retained using an external magnet and was washed thrice with ethanol, dried and stored. The organic phase of the reaction mixture was extracted with diethyl ether (3 × 20 ml) and the combined organics was washed with a % w/v Na$_2$CO$_3$ solution (20 ml) and brine solution (10 ml) consecutively and dried over anhydrous Na$_2$SO$_4$. The organic phase was filtered and concentrated under reduced pressure. The concentrated crude product was dissolved in diethyl ether and was subjected to GC–MS for characterization.

## 4.6. Homocoupling of phenylboronic acid

In a round-bottomed flask, phenylboronic acid (2.0 mmol) was mixed with 10 ml ethanol and stirred for 15 min until all solids dissolved. To this mixture, 5.0 ml of Na$_2$CO$_3$ (10% w/v) was added followed by 100 mg of Au$_{50}$Pd$_{50}$@AMNPs. The nanocatalyst was suspended in solution using ultrasonication for 30 min. The solution was heated to reflux and reacted for 4 h under air. The mixture was cooled to room temperature and diluted with distilled water (5 ml) and diethyl ether (20 ml). The catalyst was magnetically retained using an external magnet and was washed thrice with ethanol, dried and stored. The organic phase of the reaction mixture was extracted with diethyl ether (3 × 20 ml) and the combined organics was washed with a % w/v Na$_2$CO$_3$ solution (20 ml) and brine solution (10 ml) consecutively and dried over anhydrous Na$_2$SO$_4$. The organic phase was filtered and concentrated under reduced pressure. The concentrated crude product was dissolved in diethyl ether and was subjected to GC–MS for characterization.

## 4.7. Oxidation of phenylboronic acid to phenol

In a round-bottomed flask, phenylboronic acid (2.0 mmol) was mixed with 10 ml ethanol and stirred for 15 min until all solids dissolved. To this mixture, 5.0 ml of $Na_2CO_3$ (10% w/v) was added followed by 100 mg of $Au_{50}Pd_{50}$@AMNPs. The nanocatalyst was suspended in solution using ultrasonication for 30 min. The solution was heated to reflux and reacted for 4 h under pure $O_2$ gas. The mixture was cooled to room temperature and diluted with distilled water (5 ml) and diethyl ether (20 ml). The catalyst was magnetically retained using an external magnet and was washed thrice with ethanol, dried and stored. The organic phase of the reaction mixture was extracted with diethyl ether ($3 \times$ 20 ml) and the combined organics was washed with a % w/v $Na_2CO_3$ solution (20 ml) and brine solution (10 ml) consecutively and dried over anhydrous $Na_2SO_4$. The organic phase was filtered and concentrated under reduced pressure. The concentrated crude product was dissolved in diethyl ether and was subjected to GC–MS for characterization.

## 4.8. Ullmann homocoupling of 4-iodotoluene

In a round-bottomed flask, 4-iodotoluene (2.0 mmol) was mixed with 10 ml ACN and stirred for 15 min until all solids dissolved. To this mixture, 5.0 ml of $Na_3PO_4$ (10% w/v) was added followed by 100 mg of $Au_{50}Pd_{50}$@AMNPs. The nanocatalyst was suspended in solution using ultrasonication for 30 min. The solution was heated to reflux and reacted for 4 h under a $N_2$ balloon. The mixture was cooled to room temperature and diluted with distilled water (5 ml) and diethyl ether (20 ml). The catalyst was magnetically retained using an external magnet and was washed thrice with ethanol, dried and stored. The organic phase of the reaction mixture was extracted with diethyl ether ($3 \times 20$ ml) and the combined organics was washed with a % w/v $Na_2CO_3$ solution (20 ml) and brine solution (10 ml) consecutively and dried over anhydrous $Na_2SO_4$. The organic phase was filtered and concentrated under reduced pressure. The concentrated crude product was dissolved in diethyl ether and was subjected to GC–MS for characterization.

## 4.9. Hydrogenation of 1-chloro-3-nitrobenzene

In a round-bottomed flask, 1-chloro-3-nitrobenzene (2.0 mmol) was mixed with 10 ml ethanol and stirred for 15 min until all solids dissolved. To this mixture, 100 mg of $Au_{50}Pd_{50}$@AMNPs was added and suspended in solution using ultrasonication for 30 min. The solution was heated to reflux and reacted for 4 h under $H_2$ gas. The mixture was cooled to room temperature and diluted with distilled water (5 ml) and diethyl ether (20 ml). The catalyst was magnetically retained using an external magnet and was washed thrice with ethanol, dried and stored. The organic phase of the reaction mixture was extracted with diethyl ether ($3 \times 20$ ml) and the combined organics was washed with a % w/v $Na_2CO_3$ solution (20 ml) and brine solution (10 ml) consecutively and dried over anhydrous $Na_2SO_4$. The organic phase was filtered and concentrated under reduced pressure. The concentrated crude product was dissolved in diethyl ether and was subjected to GC–MS for characterization.

Data accessibility. The datasets supporting this article have been uploaded as part of the electronic supplementary material.

Authors' contributions. R.M.D.S., K.M.N.D.S. and C.G. conceived the idea and designed the project. C.G. carried out the experiments and wrote the first draft of the paper. M.S. and R.R. contributed towards characterization of materials. All authors discussed the results and commented on the final manuscript.

Competing interests. We declare we have no competing interests.

Funding. This project was not funded by any organization. Initial funding to purchase the chemicals were provided by Dr Satish Goonesinghe and the other expenses are met by the Department of Chemistry, University of Colombo and Sri Lanka Institute of Nanotechnology.

Acknowledgements. The authors acknowledge the Sri Lanka Institute of Nanotechnology (SLINTEC) and Dr Satish K. Goonesinghe for their support.

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
