## [Reviewer comments · Royal Society Open Science]

Review History

RSOS-200916.R0 (Original submission)

Review form: Reviewer 1

Is the manuscript scientifically sound in its present form?

Yes

Are the interpretations and conclusions justified by the results?

Yes

Is the language acceptable?

Yes

Do you have any ethical concerns with this paper?

No

Have you any concerns about statistical analyses in this paper?

No

Recommendation?

Accept as is

Comments to the Author(s)

The Manuscript by M. de Silva and coworkers describes the stabilization of AuPd nanoparticles and their application in the C-C coupling reaction. In my opinion, the manuscript is well written and organized. In my opinion, this manuscript is acceptable in the present form in RSOS.

Review form: Reviewer 2

Is the manuscript scientifically sound in its present form?

No

Are the interpretations and conclusions justified by the results?

No

Is the language acceptable?

Yes

Do you have any ethical concerns with this paper?

No

Have you any concerns about statistical analyses in this paper?

No

Recommendation?

Major revision is needed (please make suggestions in comments)

Comments to the Author(s)

The authors describe the preparation of the magnetically-retrievable Au/Pd nanocatalyst system, which is applied to several reactions including Suzuki coupling and Ullmann coupling reaction. The results contain several new aspects, therefore, the manuscript is worth considering for publication. However, the current version includes serious problems, so the authors need to revise it more carefully.

First in general, there are tremendous amount of similar "magnetically-reusable" catalyst systems, and too few examples referred in this manuscript. And so many previous reports and knowledges ignored, particularly about the understanding the mechanism. For example, most of the works in this manuscript almost overlaps with Sakurai's works, but only a few reports are cited and several important papers are missing, which causes the misunderstanding the analysis. The followings are the comments for the precise contents.

1) Characterization of the catalyst is not sufficient. Is it difficult to measure TEM to confirm the size and the morphology of the nanocatalysts? Also the referee could not get the precise data of XRD at M(111) region. The authors could not discuss about the system contains an alloy structure or the simply segregated structure. At least, from the results of the catalytic reactions, there is no strong evidence to form the bimetallic alloy structure.

2) The result in Table 1 (Page 5) also makes the nanocatalyst structure suspicious. These results strongly indicate that the authors fail the homogeneous reduction conditions for the formation of the nanocatalyst. Why the loading efficiency of Pd and Au is so different?

3) Page 5, l. 28: "we conclude that gold stabilizes the palladium atom". It sounds very strange and the cited papers may not express such a "stabilization effect".

4) Page 5 Table 2: H-C coupling may indicate the "homo-coupling" of Ar-I, but the readers possible misunderstand the "protonated" product, and also be confused "which homo-coupling product either boronic acid or Ar-I". Please change the naming. In addition, are there no product derived from homocoupling of PhB(OH)₂? (Most of the case, homocoupling of PhB(OH)₂ might be observed unless the seriously deoxygenated conditions)

5) Now the real active species of Suzuki-coupling reaction is NOT the surface Pd, BUT the leached-out atomic Pd (for example, *Appl. Organometal. Chem.* 2008, 22, 288; *Chem. Asian J.* 2015, 10, 2669.) The authors' explanation for the results in Table 2 may be incorrect; simply the solvent effect and the base effect lie in the concentration of the leached-out Pd species. (In addition, Ar-I is known to cap the Au surface and deactivate the catalytic reaction at lower temperature. See also *Chem. Commun.* 2013, 49, 2542.)

For understanding the mechanism, TEM measurements before and after the catalytic reaction are very helpful.

6) Page 7, Table 4: It is very difficult to accept the explanation why entries 4 and 5 gave low yield because the activation energy of the oxidative addition of these aryl bromides must be significantly low in comparison with the reaction conditions (CH₃CN reflux). There must have another reason. What are side products?

7) Page 7, Table 5 entries 1-2: For the homocoupling and/or oxygenation of PhB(OH)₂ has been comprehensively discussed (*Asian J. Org. Chem.* 2014, 3, 668.) Significant solvent effect is also reported recently (*Bull. Chem. Soc. Jpn.* 2020, DOI: 10.1246/bcsj.20200112). If such a drastic difference between entry 1 and 2 are really observed, the authors should be responsible to elucidate the mechanism as well as to demonstrate the "O₂-pressure" dependency in the selectivity of these two reactions. Judging from the mechanism, it is very difficult to understand that these two reactions can switch exclusively only by the pressure of O₂.

Decision letter (RSOS-200916.R0)

Dear Dr De Silva:

Title: A Magnetically Retrievable Air and Moisture Stable Gold and Palladium Nanocatalyst for Efficient C-C Coupling Reactions

Manuscript ID: RSOS-200916

The editor assigned to your manuscript has now received comments from reviewers. We would like you to revise your paper in accordance with the referee and Subject Editor suggestions which can be found below (not including confidential reports to the Editor). Please note this decision does not guarantee eventual acceptance.

Please submit your revised paper before 02-Aug-2020. Please note that the revision deadline will expire at 00.00am on this date. If we do not hear from you within this time then it will be assumed that the paper has been withdrawn. In exceptional circumstances, extensions may be possible if agreed with the Editorial Office in advance. We do not allow multiple rounds of revision so we urge you to make every effort to fully address all of the comments at this stage. If deemed necessary by the Editors, your manuscript will be sent back to one or more of the original reviewers for assessment. If the original reviewers are not available we may invite new reviewers.

On behalf of the Subject Editor Professor Anthony Stace and the Associate Editor Dr Chaohua Cui.

RSC Associate Editor:
Comments to the Author:
(There are no comments.)

RSC Subject Editor:
Comments to the Author:
(There are no comments.)

Reviewers' Comments to Author:
Reviewer: 1

Comments to the Author(s)
The Manuscript by M. de Silva and coworkers describes the stabilization of AuPd nanoparticles and their application in the C-C coupling reaction. In my opinion, the manuscript is well written and organized. In my opinion, this manuscript is acceptable in the present form in RSOS.

Reviewer: 2

Comments to the Author(s)
The authors describe the preparation of the magnetically-retrievable Au/Pd nanocatalyst system, which is applied to several reactions including Suzuki coupling and Ullmann coupling reaction. The results contain several new aspects, therefore, the manuscript is worth considering for publication. However, the current version includes serious problems, so the authors need to revise it more carefully.

First in general, there are tremendous amount of similar "magnetically-reusable" catalyst systems, and too few examples referred in this manuscript. And so many previous reports and knowledges ignored, particularly about the understanding the mechanism. For example, most of

the works in this manuscript almost overlaps with Sakurai's works, but only a few reports are cited and several important papers are missing, which causes the misunderstanding the analysis. The followings are the comments for the precise contents.

1) Characterization of the catalyst is not sufficient. Is it difficult to measure TEM to confirm the size and the morphology of the nanocatalysts? Also the referee could not get the precise data of XRD at M(111) region. The authors could not discuss about the system contains an alloy structure or the simply segregated structure. At least, from the results of the catalytic reactions, there is no strong evidence to form the bimetallic alloy structure.

2) The result in Table 1 (Page 5) also makes the nanocatalyst structure suspicious. These results strongly indicate that the authors fail the homogeneous reduction conditions for the formation of the nanocatalyst. Why the loading efficiency of Pd and Au is so different?

3) Page 5, l. 28: "we conclude that gold stabilizes the palladium atom". It sounds very strange and the cited papers may not express such a "stabilization effect".

4) Page 5 Table 2: H-C coupling may indicate the "homo-coupling" of Ar-I, but the readers possible misunderstand the "protonated" product, and also be confused "which homo-coupling product either boronic acid or Ar-I". Please change the naming. In addition, are there no product derived from homocoupling of PhB(OH)₂? (Most of the case, homocoupling of PhB(OH)₂ might be observed unless the seriously deoxygenated conditions)

5) Now the real active species of Suzuki-coupling reaction is NOT the surface Pd, BUT the leached-out atomic Pd (for example, Appl. Organometal. Chem. 2008, 22, 288; Chem. Asian J. 2015, 10, 2669.) The authors' explanation for the results in Table 2 may be incorrect; simply the solvent effect and the base effect lie in the concentration of the leached-out Pd species. (In addition, Ar-I is known to cap the Au surface and deactivate the catalytic reaction at lower temperature. See also Chem. Commun. 2013, 49, 2542.)

For understanding the mechanism, TEM measurements before and after the catalytic reaction are very helpful.

6) Page 7, Table 4: It is very difficult to accept the explanation why entries 4 and 5 gave low yield because the activation energy of the oxidative addition of these aryl bromides must be significantly low in comparison with the reaction conditions (CH₃CN reflux). There must have another reason. What are side products?

7) Page 7, Table 5 entries 1-2: For the homocoupling and/or oxygenation of PhB(OH)₂ has been comprehensively discussed (Asian J. Org. Chem. 2014, 3, 668.) Significant solvent effect is also reported recently (Bull. Chem. Soc. Jpn. 2020, DOI: 10.1246/bcsj.20200112). If the such a drastic difference between entry 1 and 2 are really observed, the authors should be responsible to elucidate the mechanism as well as to demonstrate the "O₂-pressure" dependency in the selectivity of these two reactions. Judging from the mechanism, it is very difficult to understand that these two reactions can switch exclusively only by the pressure of O₂.

Author's Response to Decision Letter for (RSOS-200916.R0)

See Appendix A.

RSOS-200916.R1 (Revision)

Review form: Reviewer 3

Is the manuscript scientifically sound in its present form?

Yes

Are the interpretations and conclusions justified by the results?

Yes

Is the language acceptable?

Yes

Do you have any ethical concerns with this paper?

No

Have you any concerns about statistical analyses in this paper?

No

Recommendation?

Accept with minor revision (please list in comments)

Comments to the Author(s)

The manuscript entitled "A Magnetically Retrievable Air and Moisture Stable Gold and Palladium Nanocatalyst for Efficient C-C Coupling Reactions" reports the magnetically retrievable Au/Pd nanocatalyst with highly active in Suzuki cross-coupling. In my opinion, this manuscript is acceptable in the present form in RSOS.

But, in the reference section, it is recommended to add some recently published articles.

Decision letter (RSOS-200916.R1)

Dear Dr De Silva:

Title: A Magnetically Retrievable Air and Moisture Stable Gold and Palladium Nanocatalyst for Efficient C-C Coupling Reactions
Manuscript ID: RSOS-200916.R1

Thank you for submitting the above manuscript to Royal Society Open Science. On behalf of the Editors and the Royal Society of Chemistry, I am pleased to inform you that your manuscript will be accepted for publication in Royal Society Open Science subject to minor revision in accordance with the referee suggestions. Please find the reviewers' comments at the end of this email.

The reviewers and handling editors have recommended publication, but also suggest some minor revisions to your manuscript. Therefore, I invite you to respond to the comments and revise your manuscript.

Because the schedule for publication is very tight, it is a condition of publication that you submit the revised version of your manuscript before 22-Aug-2020. Please note that the revision deadline will expire at 00.00am on this date. If you do not think you will be able to meet this date please let me know immediately.

Kind regards,

Dr Laura Smith
Publishing Editor, Journals

On behalf of the Subject Editor Professor Anthony Stace and the Associate Editor Dr Chaohua Cui.

RSC Associate Editor:
Comments to the Author:
(There are no comments.)

RSC Subject Editor:
Comments to the Author:
(There are no comments.)

Reviewer comments to Author:
Reviewer: 3

Comments to the Author(s)
The manuscript entitled "A Magnetically Retrievable Air and Moisture Stable Gold and Palladium Nanocatalyst for Efficient C-C Coupling Reactions" reports the magnetically retrievable Au/Pd nanocatalyst with highly active in Suzuki cross-coupling. In my opinion, this manuscript is acceptable in the present form in RSOS.
But, in the reference section, it is recommended to add some recently published articles.

Author's Response to Decision Letter for (RSOS-200916.R1)

See Appendix B.

Decision letter (RSOS-200916.R2)

Dear Dr De Silva:

Title: A Magnetically Retrievable Air and Moisture Stable Gold and Palladium Nanocatalyst for Efficient C-C Coupling Reactions
Manuscript ID: RSOS-200916.R2

It is a pleasure to accept your manuscript in its current form for publication in Royal Society Open Science. The chemistry content of Royal Society Open Science is published in collaboration with the Royal Society of Chemistry.

Yours sincerely,
Dr Ellis Wilde
Publishing Editor, Journals

Royal Society of Chemistry
Thomas Graham House
Science Park, Milton Road
Cambridge, CB4 0WF

Royal Society Open Science - Chemistry Editorial Office

On behalf of the Subject Editor Professor Anthony Stace and the Associate Editor Dr Chaohua Cui.

RSC Associate Editor
Comments to the Author:
(There are no comments.)

Reviewer(s)' Comments to Author:

Appendix A

Reviewer 1

The Manuscript by M. de Silva and coworkers describes the stabilization of AuPd nanoparticles and their application in the C-C coupling reaction. In my opinion, the manuscript is well written and organized. In my opinion, this manuscript is acceptable in the present form in RSOS.	The authors would like to thank the reviewer for their positive opinion about the manuscript.
---	--

Reviewer 2

The authors describe the preparation of the magnetically-retrievable Au/Pd nanocatalyst system, which is applied to several reactions including Suzuki coupling and Ullmann coupling reaction. The results contain several new aspects, therefore, the manuscript is worth considering for publication. However, the current version includes serious problems, so the authors need to revise it more carefully.	The authors wish to thank the reviewers for their valuable recommendations and critique. We have amended the manuscript to reflect some of the recommendations by the reviewers. We have addressed the questions and concerns pointed out by the reviewers. We hope the following answers satisfactorily explains our work. It should be noted that due to the current pandemic situation we will not have access to our facilities for the foreseeable future and it will not be possible to conduct further experimental work. However, as explained below we do not believe that further work is necessitated within the scope of this study.
First in general, there are tremendous amount of similar “magnetically-reusable” catalyst systems, and too few examples referred in this manuscript. And so many previous reports and knowledges ignored, particularly about the understanding the mechanism. For example, most of the works in this manuscript almost overlaps with Sakurai’s works, but only a few reports are cited and several important papers are missing, which causes the misunderstanding the analysis.	“Magnetic recyclability” is a broad area of study in ‘heterogeneous’ catalysis. We have only referred to seminal studies pertaining to similar catalytic systems to the one reported in this manuscript. Multiple mechanistic studies, particularly by Sakurai et al. have been cited as needed in the manuscript. We hope the following explanations and amendments to the manuscript will further clarify any ambiguous aspects of this work.
1. Characterization of the catalyst is not sufficient. Is it difficult to measure TEM to confirm the size and the morphology of the nanocatalysts? Also the referee could not get the precise data of XRD at M(111) region. The authors could not discuss about the system contains an alloy structure or the simply segregated structure. At least, from the results of the catalytic reactions, there is no strong evidence to form the bimetallic alloy structure.	We could not observe any of the individual reflections for Pd or Au in the XRD for the catalyst. This was not wholly unexpected as the Au and Pd concentrations are very low in comparison to the magnetite support structure. However, a broad halo appears at around $2\theta = 37^\circ\text{-}40^\circ$ region where the Au (111) and Pd (111) reflections i.e. the most intense peaks are usually seen. This is an indirect indication of the formation of a small-scale alloy structure of Au and Pd. This observation was reported by Zecca et al. (Appl. Sci. 2019, 9(15),

	2959; https://doi.org/10.3390/app9152959). This has now been clarified in the manuscript. Nevertheless, we understand that this is not sufficient evidence to conclusively claim the formation of an amorphous nano-alloy and we have removed instances referring to a nano-alloy from the manuscript to reflect this. However, EDS surface mapping does show mixed clusters of gold and palladium on the catalyst surface. Considering the superparamagnetic nature of the iron oxide core, obtaining TEM proved difficult. Since particle size was not a concern, size approximation using SEM and surface mapping using EDS proved sufficient. While TEM will provide more information, due to the current pandemic situation we will not have access to our labs for the foreseeable future.
2. The result in Table 1 (Page 5) also makes the nanocatalyst structure suspicious. These results strongly indicate that the authors fail the homogeneous reduction conditions for the formation of the nanocatalyst. Why the loading efficiency of Pd and Au is so different?	We disagree with the assertion that we failed homogeneous reduction conditions. Gold and palladium precursors and the iron oxide are ultrasonicated for 60 minutes to achieve dispersal and excess reducing agent is added rapidly under vigorous stirring. It is unlikely that the different loading efficiencies are correlated to the reduction conditions.
3. Page 5, l. 28: “we conclude that gold stabilizes the palladium atom”. It sounds very strange and the cited papers may not express such a “stabilization effect”.	As cited in the manuscript, Sakurai et al. (https://doi.org/10.1039/C5CC04432D) reported that incorporation of gold into palladium nanoclusters stabilizes the palladium by reducing leaching. Our observations agree with Sakurai’s work. We observed that initial high concentrations of palladium results in leaching and subsequent oxidation. Hence, the washed and isolated catalyst does not contain much of the palladium (Table 1, entries 2 and 3). As the gold content increases correspondingly the palladium loading efficiency increases pointing again to the “capture and stabilization” effect reported by Sakurai. This stabilization results in the difference in Pd and Au loading efficiencies. The cited work by Li et al. (https://doi.org/10.1016/j.ijhydene.2010.04.016) gives further credence to our explanation. It should be noted that we did not expect a 100% loading efficiency as this depends on the following factors.

	 i. Number of amine moieties present of iron oxide support – This is a limiting factor in how much Au and Pd can be loaded onto the surface of the iron oxide particles. ii. Rate of addition of reducing agent – As we wanted small Au and Pd clusters on the surface, rapid addition was used. However, this inadvertently results in some reduction occurring in solution, reducing the apparent loading efficiency of the metals.
4. Page 5 Table 2: H-C coupling may indicate the “homo-coupling” of Ar-I, but the readers possible misunderstand the “protonated” product, and also be confused “which homo-coupling product either boronic acid or Ar-I”. Please change the naming. In addition, are there no product derived from homocoupling of PhB(OH)₂? (Most of the case, homocoupling of PhB(OH)₂ might be observed unless the seriously deoxygenated conditions)	We agree with this recommendation and the naming has been changed and clarified to show the homocoupling product of Ar-I. The homocoupling of PhB(OH)₂ could only be observed in reactions with significant air leakage. The solvents used were purged with dry nitrogen and the system was efficiently flushed with nitrogen prior to the reaction. Only trace amounts of biphenyl (~0.1%) could be observed using GC-MS.
5. Now the real active species of Suzuki-coupling reaction is NOT the surface Pd, BUT the leached-out atomic Pd (for example, Appl.Organometal. Chem. 2008, 22, 288; Chem. Asian J. 2015, 10, 2669.) The authors’ explanation for the results in Table 2 may be incorrect; simply the solvent effect and the base effect lie in the concentration of the leached-out Pd species. (In addition, Ar-I is known to cap the Au surface and deactivate the catalytic reaction at lower temperature. See also Chem. Commun. 2013, 49, 2542.) For understanding the mechanism, TEM measurements before and after the catalytic reaction are very helpful.	We agree that the active catalytic phase is very likely leached palladium. However, our explanations of the catalytic activity still hold true. In higher polarity solvents the leached palladium would disperse more efficiently than the low polarity solvents as shown by the high yields obtained for all palladium containing systems with ACN even with the weaker base Na₂CO₃. It well known that the increased solubilization of the ionic boronate intermediate plays a significant role in the efficiency of Suzuki cross-coupling reactions. We have appended information to the relevant paragraph to clarify that the active catalytic phase is very likely leached palladium. Reference to the catalyst has been changed to “the active catalytic phase” to reflect this. Latter sections of the manuscript have also been edited to reflect this. The inhibitor effect of Ar-I does not play a part in the catalytic activity as we show that gold does not play a catalytic role in this system. Gold atoms play

	a stabilization role in maintaining the palladium reservoir in the catalyst system. Although we believe that we have presented sufficient information on the activity of the catalyst, further experimentation using TEM is not possible at this time due to the restrictions involving the current pandemic situation.
6. Page 7, Table 4: It is very difficult to accept the explanation why entries 4 and 5 gave low yield because the activation energy of the oxidative addition of these aryl bromides must be significantly low in comparison with the reaction conditions (CH₃CN reflux). There must have another reason. What are side products?	We have not endeavored to mechanistically explain the low yields of Table 4 entries 4 and 5. However, these low yields were only observed for the aryl halide coupling partners with strongly electron donating substituents. We have only presented this observation. The only side products present were the homocoupling products of the two coupling partners in trace amounts (~0.1% by GC-MS).
7. Page 7, Table 5 entries 1-2: For the homocoupling and/or oxygenation of PhB(OH)₂ has been comprehensively discussed (Asian J. Org. Chem. 2014, 3, 668.) Significant solvent effect is also reported recently (Bull. Chem. Soc. Jpn. 2020, DOI: 10.1246/bcsj.20200112). If the such a drastic difference between entry 1 and 2 are really observed, the authors should be responsible to elucidate the mechanism as well as to demonstrate the "O₂-pressure" dependency in the selectivity of these two reactions. Judging from the mechanism, it is very difficult to understand that these two reactions can switch exclusively only by the pressure of O₂.	We agree with your comment on the reported mechanism of this reaction. However, we do not claim that the switching of selectivity between the oxidation and coupling reaction are solely dependent on the O₂ partial pressure i.e concentration. Nevertheless, under repeated experimentation this result is consistent. In previously reported work the oxidation product, phenol, has been a significant by-product. Several mechanisms involving O₂ have been proposed for this reaction (as cited in the manuscript) and these are dependent on a variety of factors including metal involved, catalyst morphology, nature of the catalyst support and particle size. Kohsari et al. reported the hydroxylation of phenylboronic acid by Pd nanoparticles in the presence of H₂O₂ (https://doi.org/10.1016/j.micromeso.2018.05.045). It is possible the mechanism in our reaction involves the formation of peroxides under high oxygen conditions. However, this will require a detailed study of mechanism of only this reaction. While such a mechanistic study involving the O₂ dependent selectivity would be an interesting exercise, it is beyond the scope of this study.

Appendix B

Response for Reviewer Comments

Reviewer comments to Author:

Reviewer: 3

Comments to the Author(s)

The manuscript entitled "A Magnetically Retrievable Air and Moisture Stable Gold and Palladium Nanocatalyst for Efficient C-C Coupling Reactions" reports the magnetically retrievable Au/Pd nanocatalyst with highly active in Suzuki cross-coupling. In my opinion, this manuscript is acceptable in the present form in RSOS. But, in the reference section, it is recommended to add some recently published articles.

Author Response

We would like to thank the reviewer for this comment and the following references have been added

13. Ehara M, Priyakumar UD. 2019 Gold- Palladium Nanocluster Catalysts for Homocoupling: Electronic Structure and Interface Dynamics. *Chem. Rec.* **19**, 947–959. (doi:10.1002/tcr.201800177)
14. McVicker R, Agarwal N, Freakley SJ, He Q, Althahban S, Taylor SH, Kiely CJ, Hutchings GJ. 2020 Low temperature selective oxidation of methane using gold-palladium colloids. *Catal. Today* **342**, 32–38. (doi:10.1016/j.cattod.2018.12.017)
15. Freakley SJ, Agarwal N, McVicker RU, Althahban S, Lewis RJ, Morgan DJ, Dimitratos N, Kiely CJ, Hutchings GJ. 2020 Gold–palladium colloids as catalysts for hydrogen peroxide synthesis, degradation and methane oxidation: effect of the PVP stabiliser. *Catal. Sci. Technol.* (doi:10.1039/D0CY00915F)
16. Bellini M *et al.* 2019 A Gold–Palladium Nanoparticle Alloy Catalyst for CO Production from CO₂ Electroreduction. *Energy Technol.* **7**, 1800859. (doi:10.1002/ente.201800859)
17. Lee KU, Byun JY, Shin HJ, Kim SH. 2020 Nanoporous gold-palladium: A binary alloy with high catalytic activity for the electro-oxidation of ethanol. *J. Alloys Compd.* **842**, 155847. (doi:10.1016/j.jallcom.2020.155847)